# Using a Non-Contact Sensor to Delineate Management Zones in Vineyards and Validation with the Rasch Model

**DOI:** 10.3390/s23229183

**Published:** 2023-11-14

**Authors:** Francisco J. Moral, Francisco J. Rebollo, João Serrano

**Affiliations:** 1Departamento de Expresión Gráfica, Escuela de Ingenierías Industriales, Universidad de Extremadura, Avenida de Elvas, s/n, 06006 Badajoz, Spain; 2Departamento de Expresión Gráfica, Escuela de Ingenierías Agrarias, Universidad de Extremadura, Carretera de Cáceres, s/n, 06007 Badajoz, Spain; frebollo@unex.es; 3MED—Mediterranean Institute for Agriculture, Environment and Development and CHANGE—Global Change and Sustainability Institute, Universidade de Évora, Pólo da Mitra, Ap. 94, 7006-554 Évora, Portugal; jmrs@uevora.pt

**Keywords:** homogeneous zones, precision viticulture, Rasch model, sensors

## Abstract

The production of high-quality wines is one of the primary goals of modern oenology. In this regard, it is known that the potential quality of a wine begins to be determined in the vineyard, where the quality of the grape, initially, and later that of the wine, will be influenced by the soil properties. Given the spatial variability of the fundamental soil properties related to the potential grape production, such as texture, soil organic matter content, or cation exchange capacity, it seems that a uniform management of a vineyard is not the most optimal way to achieve higher grape quality. In this sense, the delineation of zones with similar soil characteristics to implement site-specific management is essential, reinforcing the interest in incorporating technologies and methods to determine these homogeneous zones. A case study was conducted in a 3.3 ha vineyard located near Évora, south of Portugal. A non-contact sensor (DUALEM 1S) was used to measure soil apparent electrical conductivity (ECa) in the vineyard, and later, a kriged ECa map was generated. ECa and elevation maps were utilised to delineate homogeneous zones (management zones, MZs) in the field through a clustering process. MZs were validated using some soil properties (texture; pH; organic matter—OM; phosphorous—P_2_O_5_; potassium—K_2_O; the sum of the exchange bases—SEB; and cation exchange capacity—CEC), which were determined from 20 soil samples taken in the different MZs. Validation was also performed using Rasch measures, which were defined based on the formulation of the objective and probabilistic Rasch model, integrating the information from the aforementioned soil properties at each sampling location. The comparison of the MZs was more evident with the use of the Rasch model, as only one value was to be employed in each MZ. Finally, an additional validation was conducted using a vegetation index to consider the plant response, which was different in each MZ. The use of a non-contact sensor to measure ECa constitutes an efficient technological tool for implementing site-specific management in viticulture, which allows for the improvement of decision-making processes by considering the inherent spatial variability of the soil.

## 1. Introduction

The use of precision agriculture techniques in vineyards, known as precision viticulture, is necessary for effectively managing soil spatial variability. This becomes particularly crucial in the current context of climate change, where ensuring sustainability and the quality of wine grape production is paramount. Therefore, the adoption of methods and technologies to gather information on the key variables influencing soil spatial variability throughout vineyards is essential.

It is known that soil properties are key factors for grapevine and grape phenolic development [1,2]. The amount of available water and nutrients in soil depends on their particular characteristics, including texture, depth and fertility, biology and organic matter content [3]. The fertility of the topsoil determines the availability of nutrients to vines. Since soil fertility is usually variable, understanding its spatial pattern across a particular vineyard is crucial, as fertiliser requirements and vine growth are related to the variability of fertility.

Soil texture is a key factor in determining the amount of water a soil can hold, which is particularly important in arid regions. Soils with smaller particles, that is, clay and silt, have a larger surface area than those soils where larger particles, sand, predominate. Thus, soils can hold more water when they have a larger surface area, which is the case when they contain a significant percentage of fine particles.

Soil pH is another important property as it directly impacts the availability of nutrients to grapevines. Cation exchange capacity (CEC) determines the long-term productivity of a soil. Soil texture and CEC are often related; usually, high CEC is found in soils with high clay content. Therefore, if soil texture, pH, CEC, and primary nutrient availability are considered, a rational indication of the soil fertility potential in a soil can be obtained.

Differences in the key soil properties result in variations in soil fertility. Thus, the implementation of strategies for suitable vineyard management requires the delineation of zones with permanent soil properties where their values are similar within each zone. These subfields, where similar production can be expected within each zone, are usually referred to as management zones (MZs) [4]. Implementing site-specific management strategies in a field is based on the MZs, but the accurate delineation of MZs in a field is difficult because many factors can affect crop yield and quality. However, the main physical and chemical properties of soil have to be considered by any approach to determine MZs because productivity is highly related to the spatial patterns of soil nutrients [5,6].

Accurately understanding the spatial patterns of soil properties necessitates the collection and analysis of data across the vineyard. Traditional soil sampling methods are often impractical due to the significant time and cost required to collect numerous samples, followed by laboratory analysis. A more efficient approach involves the use of sensors, which reduces the need for extensive on-site soil sampling to describe spatial variability. Soil apparent electrical conductivity (ECa) is a valuable variable for characterizing different patterns in fields and has been widely used by researchers to map soil properties, particularly those related to ECa, such as clay content, cation exchange capacity, and various soil nutrient levels [7,8,9].

Delineating homogeneous zones in a field is difficult because many factors can interact in a complex way. But the fact that some variables responsible for variability remain stable over time, such as texture and topography, provides a better understanding of the main causes of the existing spatial patterns and, As a consequence, a better delimitation of management zones. This is the first stage in the application of site-specific crop management [10], and it can allow grape growers to consider a smart methodology based on zones of uniform vine properties for sampling [11].

Today, spatial variability across soils can be studied with high spatial resolution by means of new technologies. These new tools provide intensive georeferenced information which can be analysed according to different areas within a field [12]. In this context, the use of proximal sensing constitutes an important source of data about some soil properties [11]. There are many available sensors for the proximal monitoring of soil patterns [8]. Some of the most common sensors used in precision viticulture include the ECa sensing devices, which are coupled with GNSS receivers and mounted on mobile platforms [11]. Depending on the method of electrical conductivity measurement, there are two types of electrical conductivity sensors: non-contact or contact. The non-contact sensors (e.g., EM38 and DUALEM) work on the principle of electromagnetic induction. They are composed of a transmitter and a receiver coil, usually at opposite ends of a non-conductive bar. A magnetic field is generated, inducing an electric current in the soil, which, in turn, generates a second magnetic field proportional to its conductivity [8]. The contact sensors (e.g., Veris) utilise electrodes in the shape of coulters that make contact with the soil, generating electrical currents in the soil with an electrode and measuring the potential difference with a second receiver electrode. These sensors measure the electrical resistivity [10]. ECa is affected by many soil physical and chemical properties, including texture, soluble salts, water content, organic matter content, cation exchange capacity, and many soil nutrients [7,8].

The information obtained from any sensor has to allow management decisions to vary in different locations within the field. The correct delineation of MZs is an important task to properly manage spatial variability. There are different methods to determine MZs [13,14,15]. However, a cluster analysis algorithm is an optimum technique to properly divide a field using the information stored in a geographical information system [15]. When MZs are delineated, the final step is to validate them. For this purpose, it is necessary to have information from various soil samples collected in each of the MZs and to consider in those samples the most significant variables that may influence crop productivity. In this sense, the Rasch model serves as an appropriate tool to integrate all the variables in each of the samples and perform an overall validation of the MZs based on the various soil variables being considered.

The objectives of this study were to (i) assess the soil spatial variability of a vineyard through an ECa survey with a non-contact sensor, (ii) delineate MZs based on ECa and topography data, and (iii) validate the MZs using some soil properties and the Rasch measures and utilise a vegetation index to analyse the plant response in each MZ.

## 2. Materials and Methods

### 2.1. Site Description

The field was an experimental vineyard located at the Mitra farm (University of Evora) in southern Portugal (38.532° N; 8.015° W). The vineyard is divided into two areas (Figure 1): the smaller one has 1.3 ha and is planted with white grape varieties (“AntãoVaz”, “Arinto” and “Síria”); the larger one has 2 ha and is planted with red grape varieties (“Trincadeira”, “Aragonez” and “Castelão”).

This Mediterranean region has a climate type which is classified as Csa, according to the Köppen–Geiger classification [16]. Its main characteristics are low precipitation, with a mean value lower than 600 mm, high interannual variability and practically no precipitation during summer, and a wide temperature range, with a minimum of around 0 ºC in winter and more than 40 °C (maximum) in summer.

In the experimental field, the predominant soil type is classified as a Cambisol, derived from granite [17]. These soils are characterised by slight or moderate weathering of parent material and by absence of appreciable quantities of organic matter, aluminium and/or iron compounds and illuviated clay. Usually, acid Cambisols are not very fertile, being mainly used for mixed arable farming or grazing and forest land. There are also hydromorphic soils (Leptosols and Luvisols) in the field. While Luvisols show a pedogenetic differentiation of clay content, that is, lower clay content in the topsoil and higher clay content in the subsoil, Leptosols have a very shallow horizon, limiting root growth. Soil depth in the vineyards is very variable, ranging from 0.5 to 1 m approximately.

### 2.2. Soil Apparent Electrical Conductivity (ECa) Survey and Delineation of Management Zones (MZ)

A DUALEM 1S non-contact sensor (Dualem, Inc., Milton, ON, Canada) equipped with a GPS antenna was used to measure ECa in the experimental vineyard. This sensor has a 1 m separation between its transmitter and two receivers, enabling measurements at depths of 0–0.50 m and 0–1.50 m with successive passes across the field. A four-wheel vehicle equipped with a 3 m long tow arm was built from polyvinyl chloride (PVC) and pulled by an all-terrain vehicle at an average and constant speed of 5 km h^−1^. ECa readings were effectively obtained from depths of 0–0.30 m and 0–1.30 m, taking into account that the ECa of the air is zero and considering a vehicle height of 0.20 m. The sensor was programmed to record measurements every second. ECa survey was conducted on the 24 October 2022. On this day, the soil water content had a mean value of 17.7% in the white grape vineyard and 16.2% in the red grapes vineyard. The ECa measurements have a within-field temporal stability; that is, the ECa spatial pattern is maintained over time, so the ECa survey should be conducted once in the medium term [18].

As metal objects strongly influence the electromagnetic signal, some fundamental precautions were considered with the aim of minimising or eliminating interference with the instrument, including that the sensor was mounted using a non-conductive material, the cables were placed as far away as possible from the coils, the sensor was located behind the vehicle at a distance of 3 m, and it was avoided traversing near other conductive objects.

After conducting the ECa surveys, the database was revised and erroneous values, such as negative or excessively high values, were removed. Finally, 4937 ECa values were considered, 1596 in the subparcel of the white grapes and 3341 in the subparcel of the red grapes (Figure 1). According to Serrano et al. [19], the use of the DUALEM sensor is recommended for soils with sparse vegetation cover, similar to the conditions in the experimental vineyard.

From the ECa data, kriged ECa maps were generated in each subfield utilising the extension Geostatistical Analyst of ArcGIS (version 10.5, ESRI, Inc., Redlands, CA, USA). ECa at unsampled locations was estimated with the ordinary kriging algorithm, taking into account the spatial correlation structure described with the variograms. Subsequently, the spatial patterns of ECa were visualised with the ArcMap module of the ArcGIS.

The use of geostatistical algorithms for interpolation implies that a stochastic surface describes the spatial variation of any continuous variable, ECa, in this case study. The chosen model for the trend of the random function differentiates all types of kriging [20]. When the ordinary kriging method is chosen, the trend is unknown and considered to fluctuate locally, with stationarity within the local neighbourhood.

Considering that each datum, Z(x_i_), has an associated weight, w_i_(x), and m(x) and m(x_i_) are the expected values of Z*(x) and Z(x_i_), respectively, the linear regression estimator Z*(x) is the basis of all geostatistical estimators:$$Z^{*}(x)-m(x)=\sum_{i=1}^{n}w_{i}(x)\cdot [Z(x_{i})-m(x_{i})]$$

At any unsampled location, when an estimate is generated, the weights are computed by solving a system of linear equations where the spatial variability of the analysed variable is controlled by one theoretical variogram previously fitted to the points of the experimental variogram. The weights must be computed to minimise the estimation variance, Var[Z*(x) − Z(x)], while ensuring the unbiasedness of the estimator, E[Z*(x) − Z(x)] = 0 [20,21].

Using the GPS instrument, a topographic survey of the subparcels was also performed. Consequently, elevation data were sampled with the GPS assembled on the all-terrain vehicle, and then, the digital elevation model surface (Figure 1) was generated with the triangulated irregular network (TIN) interpolation tool within ArcGIS. This vector information was later converted into a grid surface with the Spatial Analyst Tools in ArcGIS.

Homogeneous zones (management zones) were delineated using a clustering technique. As previous studies have shown (for instance, [22]), topography is a significant factor influencing the delineation of these zones. Therefore, both topography and ECa data were considered in the zone delineation process. The final classified map was generated using an unsupervised classification technique, the ISO Cluster algorithm in ArcGIS. This method organises the data in the input raster into a user-defined number of groups to produce signatures, which are used to classify the data using the Maximum Likelihood Classifier (MLC) function. Thus, the number of groups (MZs) was fixed at three in this study to ensure practicality and simplicity, as few homogeneous zones should be delineated.

### 2.3. Soil Sample Collection and Analysis—Multispectral Measurements by Remote Sensing

A total of twenty samples, each georeferenced with a GPS, were collected from both subparcels. These sample locations were chosen based on the previously defined MZ. Composite soil samples were created, consisting of nine subsamples gathered around each main sampling point. All samples were collected in October 2022. The soil samples were taken using a gouge auger and a hammer from a depth of 0–0.30 m.

The soil samples were kept in plastic bags, air-dried and analysed for their particle-size distributions (using a Sedigraph 5100, Micromeritics, Norcross, GA, USA) after passing the fine components through a 2 mm sieve. Afterwards, the fine components were analysed using standard methods [23]: pH in a 1:2.5 (soil:water) suspension, organic matter (OM), phosphorous (P_2_O_5_), potassium (K_2_O), sum of the exchange bases (SEB; calcium—Ca^2+^; magnesium—Mg^2+^; sodium—Na^+^; and potassium—K^+^) and cation exchange capacity (CEC).

Sentinel-2 optical images, obtained through the electronic platform “http://agromap.agroinsider360.com” (accessed on 15 September 2022), were used to monitor the spatial variability of vineyard vegetative vigour. The normalised difference vegetation index, NDVI, was computed from the Sentinel-2 band 4 (B4, 665 nm, 10 m spatial resolution), band 8 (B8, 842 nm, 10 m spatial resolution), band 8A (B8A, 865 nm, 20 m spatial resolution) and band 11 (B11, 1610 nm, 20 m spatial resolution). During the greatest vegetative vigour of the vineyard, that is, from May until August, four days without clouds were considered in 2022: 24 May, 28 June, 28 July and 27 August.

### 2.4. The Rasch Model—Validation of MZ

In order to validate the MZs, information obtained from the sampling locations was utilised. As multiple soil properties were measured in each sample, they were integrated into a unified measure using the Rasch model.

In the sense that the production potential is influenced by soil fertility, this model, as a measuring technique, is an innovative approach to estimating soil fertility. The objective of this study is to transcend the individual measurements related to various soil properties taken at different locations and condense them into a single, comprehensive variable. By utilising an adimensional common referent known as the latent variable, a synthesised representation of the various items can be generated. In this study, the latent variable was defined as “soil fertility”.

The Rasch model is a single-parameter model; that is, there is only one measurement parameter, which corresponds to a single dimension on a single scale to measure the classification of both the considered items (soil properties) and the subjects (soil samples). This model can consolidate some heterogeneous measures of soil properties, with different units of measurement, into an overall variable that facilitates the interpretation of potential fertility in agricultural soil. It is based on the simple idea that some items are more important to subjects than other items. Consequently, a line of measurement is constructed with the items located hierarchically according to their importance to subjects.

The categorisation of data is the first stage to achieve an adimensional measure. Thus, soil properties are described in terms of uniform rating categories; independent scale quantities can be expressed as common ratings ranging from low to high. In this case study, five categories were established for each soil property, following a methodology similar to that used in other studies [24]. Category 1 represents the lowest contribution to soil fertility, while category 5 represents the highest contribution to soil fertility. The sum of the ratings for each soil property at each sampling location results in an initial classification of all samples.

If n are the locations in the vineyard where measurements of each soil property, i, were performed, Xni is defined as the latent variable, soil fertility. Data are arranged in matrix form: the rows are the sampling locations, and the columns are the soil properties. Each cell has a value of Xni, which corresponds to the category. The sum of the item ratings is used by the Rasch model as a starting point for estimating the Rasch measures.

The probability that location n has the influence of the soil property corresponding to item i, given the parameters Bn and di, is
$$P[\mathrm{Xni}=1;\;\mathrm{Bn},\mathrm{di}]=\frac{e^{\left(B_{n}-d_{i}\right)}}{1+e^{\left(B_{n}-d_{i}\right)}}$$

As a consequence, given both parameters, Bn and di, this is the probability that location n has the influence of the soil property corresponding to item i. Although it is the formulation for a dichotomous model, it has been extended to polytomous models with many categories. Taking logarithms, log (P/(1 − P)) = Bn − di, where P = P[Xni = 1; Bn,di]; log (P/(1 − P)) is the logit of P. Larger positive values are computed for locations where pasture soil fertility is higher.

Rasch measurement construction applies a stochastic Guttman model, converting rating scale observations into linear measures. As a consequence, linear statistics can be usefully applied, and additionally, tests for goodness-of-fit to validate its item calibrations and subject measures can also be used. It is important to note that when the Rasch model is employed, the model is provided, and then the data should fit the model.

To assess the contribution of each soil property to the measurement of soil fertility in the experimental vineyard, chi-square fit statistics, the Infit and Outfit Mean Square (Infit and Outfit MNSQ), should be calculated. It is important to note that Infit and Outfit MNSQ values within the range of 0.6 to 1.5, as recommended by Bond and Fox [25], indicate an acceptable fit for an item.

The Rasch model was implemented by the Winsteps v. 4.0 computer program [26]. The mathematical formulation of this model can be revised, for example, in Ferrari and Salini [27] and Edwards and Alcock [28].

In this study, the Infit and Outfit MNSQ values (not shown) were found to fall within the acceptable range for these statistics. Therefore, all the soil properties considered were included and integrated into the characterisation of the latent variable, soil fertility. Consequently, Rasch measures for soil fertility were obtained for all sampling locations where soil samples were collected, taking into account information from nine soil properties. The procedure used in this study to determine soil fertility through the Rasch model formulation aligns with methodologies employed in previous studies [24].

The validation of the MZs was performed by computing the differences in the values for some soil properties within each zone. Later, the Rasch measures were also used to validate the MZs. To do this, the Kruskal–Wallis nonparametric test and the Dunn test as a post hoc analysis were conducted using the IBM SPSS statistical package (version 24, IBM Corp., Armonk, NY, USA). Finally, NDVI values were also used to analyse the differences in vineyard vegetative vigour between MZs.

## 3. Results

### 3.1. Delineation of Management Zones Based on ECa and Topography

The exploratory analysis, that is, the first phase of the geostatistical study, confirmed the normality of ECa data. Mean and median values were very similar (46.8 and 46.4, respectively), and furthermore, a low skewness value (0.5) and a coefficient of kurtosis close to 3 were also obtained. According to Goovaerts [21], normality is not a prerequisite for kriging, but it is a desirable property because the best absolute estimates will be generated by kriging if the random function fits a normal distribution. Moreover, the coefficient of variation was high, 58%, denoting the importance of the spatial variability of ECa in the experimental fields.

During the second phase of the geostatistical study, the spatial distribution of ECa was quantified with the variogram. Isotropic conditions were assumed because the anisotropy ratio, that is, the quotient between the ranges of the directional variograms in the major and minor anisotropic directions, was low (1.44). The best fit to the experimental variogram was provided by a theoretical spherical model (Figure 2) with the parameters range = 130.4 m; sill = 0.73; nugget effect = 0.19. As a consequence, the ratio of nugget to sill was 26%, indicating a strong spatial dependence as the ratio was around 25% [21].

Before producing the kriged map from ECa estimates at unsampled locations, a validation process was performed to provide an idea about the reliability of the final surface. Some prediction error statistics were used as diagnostics to indicate whether the kriged surface was appropriate. The root mean square error was 8.57, the mean standard error was 2.69, and the mean standardised error was 0.11. Since all these statistics were low, the kriged map was reasonable. Moreover, the fact that the standard deviation of the measured values was 27.13, higher than the root mean square error, denotes effective prediction throughout the field.

Finally, the spatial distribution of ECa in both subparcels was visualised from the kriged map (Figure 3). In the subparcel with red grape varieties, a greater degree of ECa variability was observed, particularly in the central section of the vineyard, where high ECa values predominated. In contrast, the subparcel with white grape varieties exhibited low ECa values in one part of the vineyard, characterised by a uniform pattern, while the other part showed higher ECa values with greater spatial variability.

It is well-established that ECa serves as a reliable indicator of soil variability [5], given its association with certain permanent soil properties such as clay and sand content, as well as soil depth [9,14]. Furthermore, soil moisture and nutrient availability are closely linked to the spatial variability of these permanent soil properties and, consequently, to ECa.

Vineyard responses are influenced by the permanent spatial variability of the soil, which can be delineated using ECa maps and topography (the latter influencing solar exposure and microclimate), in addition to seasonal variations in temperature, precipitation, and vine cultivars. It is worth noting that in the experimental vineyard, the ECa patterns exhibit temporal stability (as reported by Serrano et al. [5]), with values changing throughout the year in response to soil moisture conditions. When delineating the spatial variability of a particular soil, it is important to conduct the ECa survey when the soil is not excessively dry, which often occurs during the summer in the region where this case study’s vineyard is located.

The MZ map (Figure 4) was generated using both ECa and elevation maps. In the white grape vineyard, two MZs were delineated, while in the red grape vineyard, three MZs were established. Although, initially, there was an intention to define MZs in the white grape vineyard, the resulting zonation hardly differed from the zoning considering two MZs.

### 3.2. Validation of Management Zones

Usually, the validation of MZ involves comparing soil samples taken from different areas within the experimental field. Soil properties analysed in each sample are compared to assess whether the delineated zones exhibit significant differences. Soil properties analysed in each sample are compared to assess whether the delineated zones exhibit significant differences. This process requires making numerous comparisons between MZs equal to the number of soil properties considered. Essentially, this analysis determines in which MZs there are differences, and if these differences are observed across a significant number of soil properties, it can be concluded that the zones are distinct. However, when using the Rasch measures determined at each sampling location, only one comparison between MZs needs to be conducted.

To compare both validations, a total of 20 georeferenced soil samples were collected in this case study. The distribution of samples across MZs was proportional to their respective areas: seven in the less productive, nine in the intermediate and four in the more productive (Figure 4).

Table 1 shows the differences in the mean values of the soil properties among the MZs. It is apparent that most of the soil properties exhibited significant differences across the MZs. In the more productive zones, lower values of sand and higher values of clay, silt, OM, pH, SEB and CEC were observed. With respect to the assimilable phosphorous and potassium, there was a notable contrast between the MZs. A recent study in the same vineyards found a high variability in both soil properties, with coefficients of variations between 38% and 62% [5]. However, despite this, the MZs displayed significant differences in terms of soil texture and other fundamental soil properties, thus confirming the homogeneity of the delineated MZs.

When the soil data were processed by the Winsteps program, the initial analysis focused on assessing how well the data fit the model. Since the mean Infit and Outfit MNSQ values were very close to 1 (the expected value), items (soil properties) properly fit the model [29]. As a consequence, each soil property fits the general pattern of the model and, additionally, contributes to support the underlying latent variable, soil fertility.

The sum of points of all categories (raw score) and the measured values obtained from the raw scores for each soil property at each sampling location are shown in Table 2. The sampling locations are displayed in this table from the site with the highest soil fertility (highest measure) to the one with lowest soil fertility (lowest measure). Figure 3 visually shows that soil samples with higher Rasch measures are located within the MZs, where higher productivity is expected. When the mean values of these samples were analysed based on their location within each MZ, significant differences were observed between the three MZs (Table 2). Soil fertility in the most productive zone was higher than in the intermediate zone, and the least productive zone exhibited the lowest soil fertility.

When the Rasch measures were analysed for each MZ but for each individual vineyard, the differences were equally evident. In the subparcel with red grape varieties, the mean value of the two samples within the less productive zone was −3.52; for the three samples within the intermediate zone, it was −1.54; and for the three samples within the most productive zone, it was −0.61. In the subparcel with white grape varieties, the mean value of the Rasch measure for the seven samples within the intermediate zone was −0.07, and for the five samples within the less productive zone, it was −1.24. In addition to the differences in soil fertility within each of the MZs, these results also highlight variations in fertility levels in both subparcels. Nevertheless, in any case, the Rasch measures are capable of distinguishing zones that should be differently managed.

With respect to the vineyard vegetative vigour measured by NDVI, Table 3 shows that as the vegetative cycle draws on, the NDVI decreases, and higher NDVI values were in MZ with more productive potential.

## 4. Discussion

According to Ammoniaci et al. [7], precision viticulture currently focuses on four key areas: the evaluation of spatial variability, the delineation of MZs, the development of variable rate technologies and the implementation of site-specific management. The first two areas, the analysis of soil spatial variability and the delineation and validation of MZs, were addressed in this study.

Mobile soil ECa measurements are the most efficient method for mapping soil spatial variability, saving time and resources not only during the planning and establishment of new vineyards but also in the management of established ones. In this study, the high spatial variability of ECa was initially evidenced by its high coefficient of variation, 59.95%. This suggested the existing soil spatial variability in the vineyard and the convenience of delineating MZs.

As in similar studies [5,18], soil elevation was also incorporated into the process of delineating MZs in this vineyard. It is known the important influence of elevation on crop development, water accumulation and flow, and, in turn, redistribution of organic matter and soil nutrients is closely related to elevation [30]. However, given the absence of substantial elevation differences and excessively steep areas within the experimental vineyard, topography had a limited impact on the delineation of the MZs. Consequently, the ECa and MZ maps exhibited similar patterns in this study, suggesting that the ECa map alone could effectively define the MZs.

The MZs in each subparcel had different shapes. In the red grape vineyard, the MZs were arranged concentrically. Moreover, the more productive zone was subdivided. From a practical perspective for site-specific management, it would be advisable to define a continuous zone in this area. Conversely, in the white grape vineyard, both MZs were more compact. Considering site-specific management, this vineyard can be divided into two areas, with the division aligning along the north–south limit of both MZs.

Although the validation of the MZs was initially conducted using soil variables measured in field-collected samples located through targeted sampling considering the MZs, a second validation was performed using Rasch measures. Both validations confirmed that the MZs were distinct. However, the Rasch model formulation provided a single value for each MZ, simplifying the comparison process. Consequently, the MZ with more productive potential has a soil profile with a higher Rasch measure (indicating higher soil fertility). This profile includes higher clay content, pH, OM, SEB and CEC, along with lower sand content.

There are numerous studies where relationships between ECa and soil texture have been established [6,7,9]. These studies often reveal a positive correlation between ECa and clay content and a negative correlation with sand content. Moreover, other soil properties are also related to ECa, such as CEC, pH, organic matter and different soil nutrients [5,6,7,31]. In this case study, the proposed zoning explained variations in some key soil properties related to soil fertility, such as clay content, CEC or OM. However, the deviation of assimilable phosphorus and potassium from the expected trends, with higher values in the MZ with lower productivity potential, may be attributed to local factors, such as the proximity of a pigsty near the vineyard. Nevertheless, the low values of assimilable phosphorus and potassium in the most productive MZs, around 30–40 mg kg^−1^, indicate higher nutrient extractions and, additionally, highlight the importance of considering spatial variability in vineyard management.

Using remote sensing indices, such as NDVI, could also be useful for categorising zones in vineyards as, for instance, Ferrer et al. [32] and Tardaguila et al. [10] suggested. Hubbard et al. [33] obtained significant correlations between NDVI and ECa during the growing season, suggesting that ECa and soil texture could be good estimators of vegetation properties.

It is important to denote that further studies in other regions and soil types should be performed as this case study is only exploratory and restricted to a particular vineyard. Although a direct relationship between crop yield and NDVI could be apparent, that is, high crop yield is expected in zones where the vegetative vigour is also higher, sometimes higher quality wine is produced in zones of low vigour [34]. The production of wines of recognised quality is essential in modern oenology, and, in this sense, it is very important to establish the potential quality of wine in the vineyard. As a consequence, the impact of the soil spatial variability on productivity (using yield maps) and on grape quality (wine quality) will be evaluated in the future, clarifying the agronomic impact of the MZ delineation.

## 5. Conclusions

The delineation of MZs in a vineyard can be effectively achieved using an ECa sensor, as the spatial variability of the main permanent soil properties (such as texture, OM, pH, the sum of the exchange bases and cation exchange capacity) are related to the ECa pattern.

After determining the MZs, it is necessary to validate them because this is the first step to implementing site-specific management in a vineyard. To perform the validation, soil information has to be obtained from some locations throughout the field. In this sense, a useful tool to integrate data from different soil variables is the formulation of the Rasch model, obtaining estimates of soil fertility at each sampling location. Utilising Rasch measures simplifies the validation process.

When new vineyards have to be designed, an accurate delineation of MZs can be relevant. They can also be useful for guiding viticultural practices and a more informed decision-making process in site-specific input management. Moreover, grape growers can implement smart soil sampling, that is, an optimised technique to take soil samples in the vineyard. Consequently, a more cost-effective field management, besides energetic and environmental benefits, can be expected using precision viticulture techniques.

## Figures and Tables

**Figure 1 sensors-23-09183-f001:**
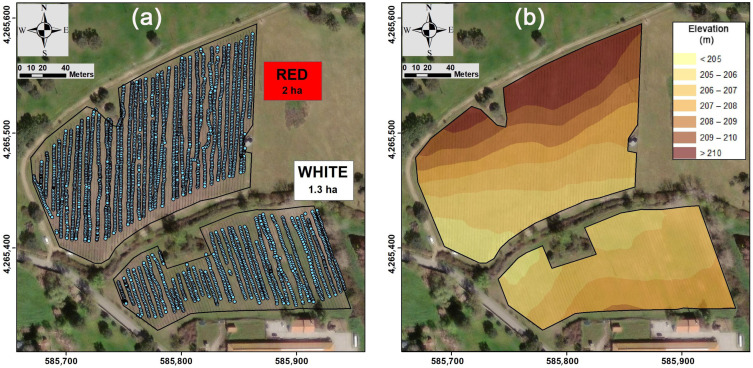
Study site: (**a**) blue dots indicate the locations where soil apparent electrical conductivity (ECa) was measured; (**b**) digital elevation model (spatial resolution = 1 m; coordinate system: ETRS89/UTM zone 29N).

**Figure 2 sensors-23-09183-f002:**
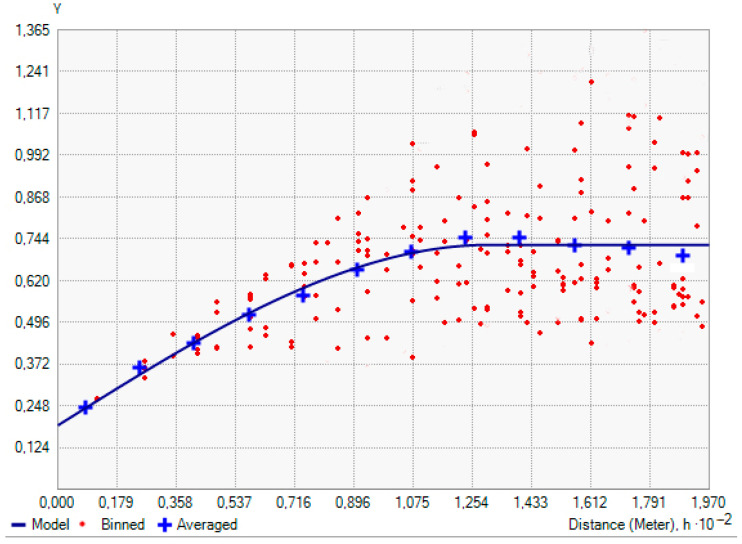
Experimental variogram (points) and theoretical spherical model (line) for soil apparent electrical conductivity (ECa).

**Figure 3 sensors-23-09183-f003:**
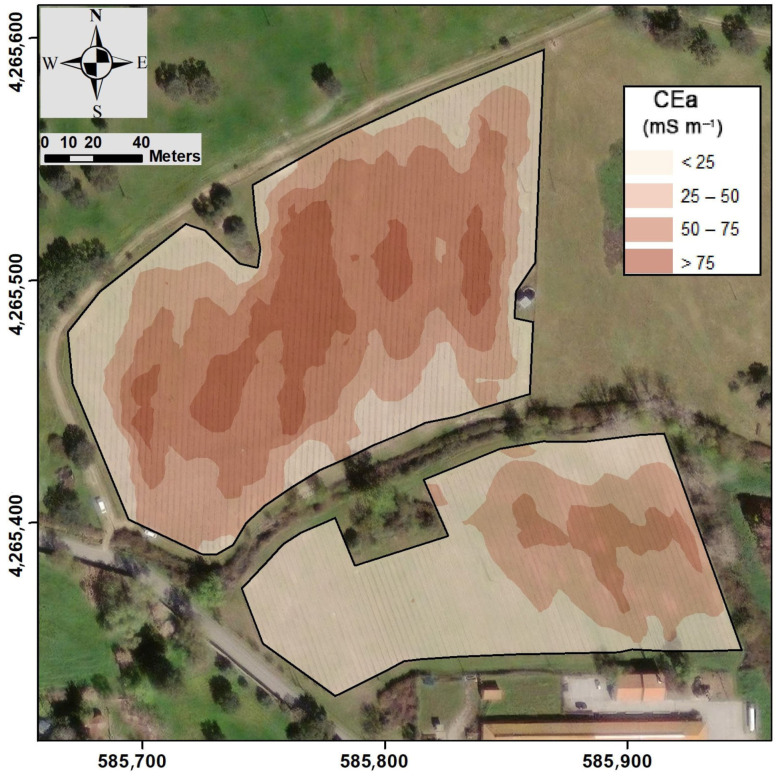
Kriged map of soil apparent electrical conductivity, ECa (spatial resolution = 1 m; coordinate system: ETRS89/UTM zone 29N).

**Figure 4 sensors-23-09183-f004:**
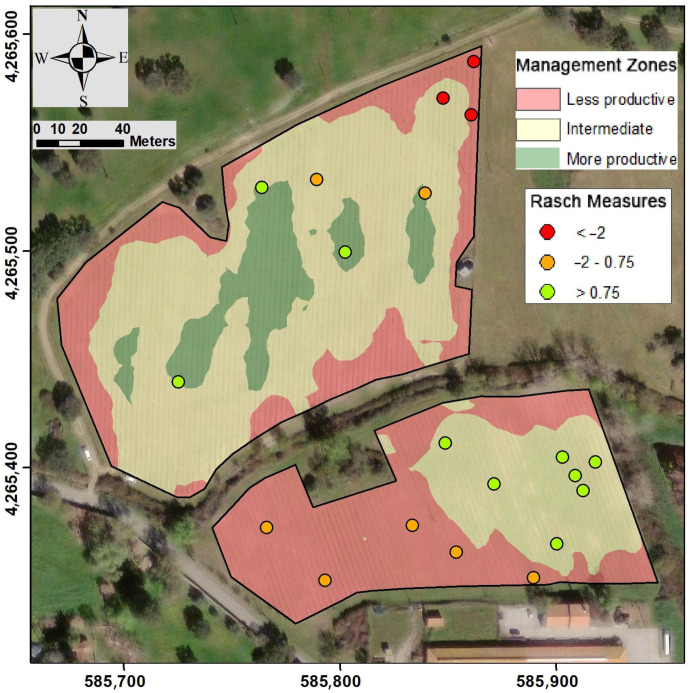
Management zones based on soil apparent electrical conductivity (ECa) and elevation. Sampling locations used for validation, with their Rasch measure, are also shown as dots (spatial resolution = 1 m; coordinate system: ETRS89/UTM zone 29N).

**Table 1 sensors-23-09183-t001:** Mean values of each soil property for samples in the less productive zone (Zone_less), the intermediate (Zone_medium) and the most productive zone (Zone_more) according to the delineation of the homogeneous zones.

	Sand (%)	Clay (%)	Silt (%)	OM (%)	pH	P_2_O_5_ (mg kg^−1^)	K_2_O (mg kg^−1^)	SEB (cmol kg^−1^)	CEC (cmol kg^−1^)
Zone_less	77.6 a	10.8 a	11.6 a	1.1 a	6.0 a	146.6 a	105.0 a	3.3 a	14.5 a
Zone_medium	70.3 b	14.1 b	15.6 b	1.8 b	6.6 b	370.9 b	97.5 b	9.5 b	15.3 b
Zone_more	68.5 c	13.1 c	18.4 c	1.2 a	6.6 b	19.9 c	33.0 c	15.0 c	21.3 c

OM—Organic matter; P_2_O_5_—Assimilable phosphorous; K_2_O—Assimilable potassium; SEB—Sum of the exchange bases; CEC—Cation exchange capacity (CEC). Different letters indicate significant differences (*p* < 0.01) according to the Dunn test.

**Table 2 sensors-23-09183-t002:** Mean values of each soil property for samples in the less productive zone (Zone_less), the intermediate (Zone_medium) and the most productive zone (Zone_more) according to the delineation of the homogeneous zones.

	Rasch Measure
Zone_less	−1.89 a
Zone_medium	−0.50 b
Zone_more	−0.59 c

Different letters indicate significant differences. (*p* < 0.01) according to the Dunn test.

**Table 3 sensors-23-09183-t003:** Mean values of the normalised difference vegetation index (NDVI) of vineyard (white and red grapes fields) on four dates in 2022, in the less productive zone (Zone_less), the intermediate (Zone_medium) and the most productive zone (Zone_more) according to the delineation of the homogeneous zones (values in the Zone_less within the red grapes field were not available).

		24 May	28 June	28 July	27 August
Red grapes	Zone_less	-	-	-	-
	Zone_medium	0.44	0.37	0.31	0.30
	Zone_more	0.50	0.44	0.34	0.33
White grapes	Zone_less	0.50	0.47	0.35	0.34
	Zone_medium	0.53	0.55	0.43	0.40

## Data Availability

The data presented in this study are available on request from the corresponding author. The data are not publicly available because they are part of a project that is ongoing (not completed).

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
