# Peer review of "Using a Non-Contact Sensor to Delineate Management Zones in Vineyards and Validation with the Rasch Model"

_sensors, 2023, doi:10.3390/s23229183_

Round 1

Reviewer 1 Report

Comments and Suggestions for Authors

Dear Editor,

I am pleased to send you my Review report below.

Article title: Using a non-contact Sensor to Delineate Management Zones in Vineyards and Validation with the Rasch Model

Manuscript ID sensors-2651621

General Comment:

This study addresses a critical topic in producing high-quality wines, highlighting the fundamental influence of soil on wine quality. The focus on the spatial variability of soil properties and the need for site-specific management to enhance grape quality is an intriguing approach.

The authors propose using the apparent soil electrical conductivity (ECa) sensor to generate an ECa map and define homogeneous management zones (MZs). This is a well-planned strategy to facilitate precision agriculture. Furthermore, using the Rasch model can add reliability to the MZs.

Overall, this study underscores the importance of considering soil variability in vineyard management and how technology and statistical models can be valuable tools in contributing to oenology and crop management.

However, the manuscript has several areas that need improvement:

The introduction could be more concise, focusing on the core theme and explaining the importance of soil structure in water availability. The writing, especially in the second paragraph of the introduction, requires correction. Additionally, details about soil depth and the expression of soil moisture content are needed. The description of geostatistical methodology is insufficient, and a more comprehensive validation is required, including a thorough analysis of the spatial variance model and cross-validation results. A description of the method for determining particle size is also missing. The classification based on the Rasch model should be supported by crop productivity measurements and consideration of important physical properties. Tables 1 and 2 require adjustments in data presentation. The importance of comparing soil fertility indicators with other productivity and crop quality indicators is emphasised, and a deeper focus on improving soil knowledge through rapid prospecting systems is suggested.

The introduction, in general, is rather lengthy and could be rewritten to be more concise, getting to the core of the matter. For example, there's no need to explain that "These sensors measure the electrical resistivity, which is the inverse of the electrical conductivity [10]." in this context. In the first paragraph of the introduction, the importance of soil structure in water availability is not mentioned. In the second paragraph, there is a sentence that is not well-phrased: "The combination of a soil (soil texture) and its acidity (pH) determines the extent to which nutrients are available to plants."

Regarding the soils, there is a lack of information about soil depth. It's unclear whether soil moisture content is expressed in gravimetric or volumetric terms, which should be specified. Surprisingly, the authors describe three types of soils (cambisols, leptosols, and lluvisols) on a very small area of 2 hectares and 1.3 hectares, respectively.

Concerning the methodology, the description of geostatistical methods and their results is insufficient. A valid geostatistical analysis needs to examine the validity of the assumptions underlying the statistical modelling for constructing spatial variability maps. The authors do not indicate whether the data are stationary or have undergone a transformation to remove trends, and they also do not analyse the data distribution. Statistical data distributions should be "reasonably" normal (consult normality tests), and if not, the need for or the impact of a transformation to normality should be discussed or indicated. It's also important to analyse if there is co-dependence with other variables, such as terrain elevation. Regarding the spatial variance model, the authors should present a graph comparing the semivariogram of the experimental data with that of the model used to create distribution maps and briefly discuss it. Finally, the authors should provide cross-validation results to determine the quality of the prediction. I recommend consulting the referenced works below:

Universidad de Santiago de Compostela, Escuela Politécnica Superior, Departamento de Ingeniería Agroforestal. Medida de la conductividad eléctrica aparente del suelo por inducción electromagnética y variabilidad espacial de propiedades físicas y químicas del suelo. Autor: Glécio Machado Siqueira. Director: Jorge Dafonte Dafonte. Octubre de 2009.

MATHERON, G. (1971) The theory of regionalised variables and its applications. Cahiers du Centre de Morphologie Mathématique de Fountoinebleau. 

VIEIRA, S.R.; HATFIELD, J.L.; NIELSEN, D.R.; BIGGAR, J.W. (1983) Geostatistical theory and application to the variability of some agronomical properties. Hilgardia, 51(3): 1-75.

In the description of the method for determining particle size distribution, they mention the procedure for disaggregating the soil before measurement with the Sedigraph. I urge the authors to reference or describe the procedure for disaggregating the primary soil particles.

The authors propose that: "The objective of this study is to transcend the individual measurements related to various soil properties taken at different locations and condense them into a single, comprehensive variable. A synthesised representation of the various items can be generated by utilising an adimensional common referent known as the latent variable. In this study, the latent variable was defined as 'soil fertility'." Despite the principles of the Rasch method for assessing fertility being established in reference number 23, the classification of MZs is purely statistical and not validated against crop productivity measures. This is why this method is based on the classification of values of soil property indicators related to fertility that are well-known. However, important physical properties that substantially affect crop performance, such as water-holding capacity, soil depth, or penetration resistance, are not considered. In my opinion, these factors should be somehow addressed in the manuscript. This would help to focus the object and scope of this study more precisely.

[23] Moral, F.J.; Rebollo, F.J. Characterization of soil fertility using the Rasch model. Journal of Soil Science and Plant Nutrition 2017, 17(2), 486–498.

Regarding sampling, in the 2-hectare plot, only 8 points were analysed for chemical determinations; in my opinion, this number is insufficient for reliable validation.

In Table 1, it is necessary to add the standard deviations of each analysed soil property. Additionally, the mean values show an excessive number of decimal places that are incongruent with the precision of the analytical methods. The statistical method used to calculate the significance level of the MZ factor should be indicated.

The classification results do not seem to align well with the Rasch measurements. For example, in the less productive zone, there are 5 points with Rasch values between -2 and 0.75. Of the 3 points with Rasch measurements less than two, only one is clearly in the less productive zone, another is in the intermediate zone, and the third is on the boundary between the less productive and intermediate zones.

In each paragraph, It is unnecessary to mention that the WinSteps program was used since it is implicitly understood that it is used, as no other program is mentioned.

In Table 2, displaying the Rasch measurement error would also be advisable.

Finally, while it is vital to establish practical soil fertility indicators that can be used for zoning the productive capacity of agricultural soils, it is essential to compare the values of these indicators with other indicators of the spatial distribution of productivity and crop quality.

The conclusions drawn in this study do not fundamentally differ from those obtained in similar studies applying this non-contact method of soil electrical conductivity and salinity measurement using the Rasch method. The only difference is that it is applied to a different type of crop, in this case, vineyards. In my opinion, this contribution has limited significance in scientific knowledge. In this regard, it could be enhanced by conducting a comparative discussion with similar studies examining the Rasch indicator's validation or correspondence with homogeneous zones defined by non-contact sensors. I believe that delving deeper into improving our understanding of soil through these rapid prospecting systems is of great interest, and therefore, the focus should be on this aspect.

Author Response

Please, see the attached document

Reviewer 2 Report

Comments and Suggestions for Authors

This manuscript sensors-2651621 investigated the use of EMI sensors for delineating management zones (MZ) within vineyard fields, and MZ was validated through the following soil sampling and analysis. While the scientific contribution of this work may not be groundbreaking, it is notable for its utilization of existing tools in a novel application. The article is well-written in proficient English and exhibits a well-structured format, though authors often use the wrong articles. Figures and tables effectively convey the intended results. However, there are several issues that should be addressed to enhance the article's clarity and readability. Given the scope of revisions required, I recommend a major revision of this manuscript.

The primary cavity of this work revolves around the conventional inquiries related to the constraints of EMI sensors. While these sensors effectively delineate zones in a field, they fall short of providing insights into the underlying reasons for variations. Similar limitations are evident in this study, as expected. The proposed methodology adeptly delineates MZ, as duly validated. However, a critical concern arises regarding the decision-making and implementation of site-specific management in vineyard fields without additional sampling or soil analyses. It is recommended that the authors address this issue as a priority. Furthermore, providing potential solutions to mitigate this challenge would enhance the overall quality of the discussion.

Some specific queries:

Abstract-

1.       Why is SOM mentioned as an alternative to the CEC?

2.       EC is referred to as an efficient technological tool, which is a wrong perception about EC, but should referred to its measuring sensor, EMI.

Introduction-

1.       What does the author mean by similar permanent soil properties? pg.2

2. The Rasch model has been not introduced properly, which makes it difficult to understand this model and its use in the current work. Pg.3

Methodology-

1.       Fig 1 is missing the Grid (spatial index).

2.       Don’t you think point kriging done here is biased/affected by the direction of scanning? Why not use block kriging instead?

3.       You are mixing the use of classification and clustering terms. Be sure that you are using clustering for zone delineation, not classification. Need a clear explanation of how the ISO cluster works since the ISO clustering method may not be very familiar to most readers, unlike k-means. Pg.4

4.       So far, I have not gotten a clear explanation of the Rasch model, how it works, what is its structure, parameters, and so on. You should provide a clear description of this Rasch model. Pg.6

Results-

1.       What do you refer to as high or low variability? What is your measure to distinguish high from low variability? If you are referring to the spatial pattern, then again what index you are referring to? Pg.6

2.       Fig 2 is missing the Grid (spatial index) like Fig 1.

3.       What do you mean by permanent spatial variability? If soil moisture affects the variability (as you also mentioned!), the question would be to what extent moisture affected the current measure of ECa. Did you take this into consideration/correct, how? Pg.7

4.       How did you come up with 2 or 3 MZ? What optimization criterion did you use?

5.       Are not you repeating the use of acronyms at this late stage?

6.       Table 1: As I understood you tabulated the mean of each soil property and indicated significant differences using letters. A core question is that how did you neutralize the impact of unequal sample size, as observed per MZ?

Comments on the Quality of English Language

Language quality is quite good and can be improved by confirming the correct use of articles.

Author Response

Please, see the attached document

Round 2

Reviewer 1 Report

Comments and Suggestions for Authors

Dear Editor,

I have reviewed the changes made by the authors in response to my previous comments. The improvements are substantial and have enhanced the quality and clarity of the article. I believe the manuscript can be published

Regards.

Author Response

Thank you for your comments and suggestions.

Reviewer 2 Report

Comments and Suggestions for Authors

The authors have taken some steps to improve their manuscript. Still, I have few observations to resolve-

1. The author did not get what I asked to explain regarding the sample’s requirements after MZ delineated by EMI sensor. I did not ask to explain their sampling design limited by budget. It requires explaining the limitation of EMI sensors and thus for this study and propose a method to overcome this sample requirement issue.

2. The author has indicated that sub-section 2.4 adequately explained this Rasch model, though I have not seen new addition in this revision. I would suggest authors to explain a bit more clearly for the readers who are not familiar with this model.

3. Fig 1:  Grid is not added yet. I asked to add grid of geo-coordinates. Actually, it should be done for all the maps.

4. My query about MZ number is not solved. They way author has selected MZ numbers is subjective, limiting this work for future reproduction of the similar work.

Comments on the Quality of English Language

Enough good to go.

Author Response

Please, see the attached document
